# Survival Benefits Based on the Number of Lymph Nodes Removed during Radical Nephroureterectomy for Upper Tract Urothelial Carcinoma: Systematic Review and Meta-Analysis

**DOI:** 10.3390/jcm9061933

**Published:** 2020-06-21

**Authors:** Min Soo Choo, Sangjun Yoo, Hyeong Dong Yuk, Chang Wook Jeong, Min Chul Cho, Cheol Kwak, Hyeon Jeong, Hyeon Hoe Kim, Ja Hyeon Ku

**Affiliations:** 1Department of Urology, Seoul National University College of Medicine, Seoul Metropolitan Government Boramae Medical Center, Seoul 07061, Korea; snuhuro@gmail.com (M.S.C.); ebend@naver.com (S.Y.); scarlet1995@hanmail.net (M.C.C.); drjeongh@gmail.com (H.J.); 2Department of Urology, Seoul National University College of Medicine, Seoul National University Hospital, Seoul 03080, Korea; armenia8@snu.ac.kr (H.D.Y.); drboss@snuh.org (C.W.J.); mdrafael@snu.ac.kr (C.K.); hhkim@snu.ac.kr (H.H.K.)

**Keywords:** lymph node dissection, upper tract urothelial carcinoma, radical nephroureterectomy, survival benefits, meta-analysis

## Abstract

The role of lymph node dissection (LND) is still controversial for upper tract urothelial carcinoma (UTUC), and there are no guidelines regarding its use. This study was conducted to find a higher level of evidence for the survival benefits based on the number of LNs removed during radical nephroureterectomy (RNUx) through a systematic review and meta-analysis. We included studies comparing patients who underwent LND during RNUx for UTUC. We searched the major electronic databases (Pubmed, Embase^®^, and Scopus^®^) and conducted manual searches of the electronically available abstracts of the major international urology cancer meetings [American Society of Clinical Oncology (ASCO), American Urological Association (AUA), and Eropean Association of Urology (EAU)] prior to April 2019 using grouped terms of nephroureterectomy (nephroureterectom*) and lymph node excision (lymphadenectomy; lymph + node*; lymph* + metasta*) with variations in the terms. Study selection, data collection, and risk of bias assessment were performed by two independent authors (A and B). Six retrospective case-control studies included a total of 33,944 patients who underwent RNUx for UTUC, 5071 of whom underwent LND and were finally included in the meta-analysis. The pooled hazard ratio (HR) in these studies revealed that an increased number of LNs removed during RNUx was associated with improved cancer-specific survival (CSS) in patients with UTUC (HR = 0.95, 95% CI: 0.91–0.99; *p* = 0.07). In addition, increased numbers of LNs removed were associated with improved overall survival (OS) in pN0 patients. However, in pN+ patients, the number of LNs removed showed no survival benefit on CSS, overall survival (OS), or progression-free survival (PFS). Higher numbers of LNs removed during RNUx were associated with improved survival outcomes in patients with UTUC. This study confirmed that LND also has oncological benefits in UTUC patients. Although still a controversial topic, meticulous LND must be considered, and efforts should be made to eliminate as many LNs as possible when administering RNUx for UTUC, especially in patients without clear evidence of LN metastasis.

## 1. Introduction

Upper tract urothelial carcinoma (UTUC) is a relatively rare neoplasm that accounts for approximately 5–10% of all urothelial carcinoma (UC), and the annual incidence in Western countries is one to two cases per 100,000 inhabitants [1,2]. However, UTUC is a highly aggressive malignancy that tends to present with lymph node (LN) metastasis in 20–40% of the patients as early as the initial diagnosis with invading musculature [3,4]. LN metastasis is a well-established poor prognostic factor with a five year survival rate of 10–30% [5].

Pelvic lymph node dissection (LND) in urothelial carcinoma of the bladder is accepted as a fundamental standard procedure for surgical management [6,7]. However, the role of LND for UTUC, which has similar histological and oncological phenotypes to bladder UC [8], is still controversial. Currently, there are no guidelines regarding its role [2]. This may be due to the difficulty of establishing evidence because of its low incidence [3]. Therefore, it is a suitable subject for a systematic review and meta-analysis.

LN metastasis in UC tends to develop in regional lymph nodes along lymphatic drainage [9]. Therefore, systemic LND can be hypothesized to have potential therapeutic benefits and provide more accurate staging information for postoperative risk stratification [10,11]. Unfortunately, there is still no consensus on the optimal standard template of LND in UTUC. Originally, hilar and abdominal para-aortic or para-caval LNs for renal pelvis tumors and intra-pelvic LNs for ureters were considered the regional LNs for UTUC based on studies from the1980s [12,13]. However, recent studies have reported that the extent of regional LNs involvement is influenced by laterality as well as the location of the primary tumor [14]. In addition, more than 30% of LN metastases were found outside the previously defined regional LN extent area [8,15]. This may suggest the need for an extended LND template for UTUC.

The number of LNs removed has been proposed as a surrogate for the anatomical extent of LND [8]. Recently, the number of LNs removed in prostate cancer as well as in bladder UC has been reported as an important predictor of oncologic outcomes [16]. However, only a small number of single-center studies have reported the significance of the number of LNs removed on clinical outcomes for UTUC. In this study, we aimed to identify higher-level evidence for survival benefits based on the number of LNs removed during radical nephroureterectomy (RNUx) through a systematic review and meta-analysis.

## 2. Methods

The purpose of this study was to investigate the prognostic effect of the number of LNs removed on patients undergoing RNUx for UTUC. We searched the results in the literature through a systematic review and tried to provide the latest evidence through meta-analysis. This study was conducted and reported according to the general guidelines recommended by the Preferred Reporting Items for Systematic Reviews and Meta-analyses (PRISMA) statement.

### 2.1. Definitions

We attempted to include all studies without specific limitations in the study design, including randomized controlled clinical trial studies and non-randomized controlled clinical trial studies, prospective cohort studies, and prospective and retrospective comparative studies. We included no restrictions on the publication status. We considered studies for inclusion that included participants who underwent RNUx for UTUC. We made no exclusions based on age or ethnicity. The number of LNs removed during RNUx was used as a comparator intervention. We planned the inclusion of any open, laparoscopic, and robotic surgical interventions. The primary outcome was overall survival (OS) according to the number of LNs removed during RNUx, and the secondary outcomes were cancer-specific survival (CSS) and progression-free survival (PFS).

### 2.2. Search Strategy

We used strict inclusion and exclusion criteria to limit heterogeneity across the studies. This study included studies comparing patients who underwent LNDs during RNUx for UTUC. We considered studies reporting OS, CSS, and PFS outcomes. We also included studies that reported the hazard ratio (HR) and corresponding 95% confidence interval (CI) for survival outcomes or studies where sufficient information about survival data in the paper could be extracted using the methods reported by Tierney et al. [17]. The number of subjects was not limited. We excluded studies on animals and those reporting potentially overlapping study populations for the same outcome. Case reports, letters, editorial, and book chapters were also excluded.

We searched the major electronic databases (PubMed, Embase^®^, and Scopus^®^) and conducted manual searches of the electronically available abstracts of the major international urology cancer meetings [American Society of Clinical Oncology (ASCO), American Urological Association (AUA), and Eropean Association of Urology (EAU)] prior to April 2019 using grouped terms of nephroureterectomy (nephroureterectom*) and lymph node excision (lymphadenectomy; lymph + node*; lymph* + metasta*) with variations in the terms. To improve the specificity of the systematic review, we used the SIGN search filter for Embase^®^ and the search filter proposed by Shojania for PubMed [18]. In addition, we manually screened the reference lists of the publications identified as potentially relevant to this review for additional studies. The ClinicalTrials.gov website was also screened to identify unpublished trials.

### 2.3. Study Selection

A list of titles and abstracts was screened by two independent authors (MSC and SY). For studies judged as relevant clinical studies in the primary screening, the fully published article was reviewed. To identify duplicate studies, we checked author names, study date, location and duration, details of the interventions, number of participants, and baseline characteristics of the potentially relevant studies. Any disagreements regarding study eligibility were resolved through discussion and consensus or by deferring to a third review author (JHK). We included an adapted PRISMA flow diagram showing the study selection process [19].

### 2.4. Data Collection

Data extraction was conducted by the same two authors (MSC and SY) independently using standardized item forms. We resolved any disagreements through discussion and consensus or by consultation with a third review author (JHK). For duplicate studies, we included the study with the most complete data set or a primary trial. We used the log HR and variance as the summary outcome measure from all trials in the meta-analysis. For each study, HR and 95% CI of the survival rate were derived to evaluate the impact of the number of LNs removed on the survival of patients with UTUC.

### 2.5. Risk of Bias Assessment

The risk of bias assessment in the included studies was also conducted by the same two authors (MSC and SY) independently. The Newcastle–Ottawa scale for case-control studies was used as a quality assessment tool. For each study, we assessed eight items in three categories: selection, comparability, and exposure. A study can be awarded a maximum of one star for each numbered item within the selection and the exposure categories. A maximum of two stars can be given for comparability. Disagreements were resolved by discussion or through a consensus reached with a third review author (JHK). The publication bias of the selected studies was detected using funnel plots, and Egger’s test was used to assess the publication bias statistically [20].

### 2.6. Statistical Analysis

For continuous data, the weighted mean difference between the intervention group and the control group was analyzed. The mean difference between the two groups was incorporated into the inverse variance method in the meta-analysis. The relative risk was calculated for the categorical data. The HRs were pooled comparing OS, CCS, and PFS for the number of LNs removed from the UTUC patients. Because HRs could differ among studies, a random-effects model was used to model potential heterogeneity. The assessment of statistical heterogeneity was analyzed and quantified by the *I*^2^ statistic. *I*^2^ > 50% and *p* < 0.1 suggested obvious heterogeneity [21]. A *p*-value of < 0.05 was considered statistically significant. Statistical analyses were carried out using Review Manager version 5.3 (The Cochrane Collaboration, Copenhagen, Denmark) and R 3.6.3 for Windows (R Development Core Team, Vienna, Austria; http://www.R-project.org).

## 3. Results

### 3.1. Literature Search and Characteristics of the Included Studies

We identified 1728 potentially relevant articles from PubMed (*n* = 332), Embase^®^ (*n* = 814), and Scopus^®^ (*n* = 582). Through the screening of conference abstracts or references of the selected studies, one additional record was identified. After removing 1009 duplicates, 694 studies were additionally excluded by careful review of the titles and abstracts. Eighty-two were not in English; 242 were animal studies; 186 were letters, editorials, or case reports; 102 were reviews or book chapters; and 82 were obviously different from the wanted paper. Based on an in-depth full-text article review, we removed a further 19 studies considered irrelevant. One study was excluded because of insufficient outcomes. Ultimately, six unique studies were included in the quantitative synthesis. Figure 1 shows the PRISMA flowchart of this systematic review.

### 3.2. Description of the Included Studies

The six retrospective case-control studies included a total of 33,944 patients who underwent RNUx for UTUC, 5071 of whom underwent LNDs and were finally included in the meta-analysis. The characteristics of the included studies are presented in Table 1. Five studies provided specific data on the pathologic T stage of the 4253 patients included. There were 2123 (49.9%) patients with localized pT1-2 disease and 2130 (50.1%) patients with advanced pT3-4 disease. There were also five studies providing specific data on the pathologic *N* stage of 4629 included patients. The incidence of LN metastasis in these studies was 28.2% (1305 patients) in the pooled data and ranged from 13.2% to 31.2% in each study. One study reported survival outcomes as OS and CSS according to the number of LNs removed [10]. Two studies presented oncologic data on CSS and PFS [22,23]. There were three papers providing survival results only for OS or CSS [24,25,26].

### 3.3. Quality of the Individual Studies

The results of the quality assessment for the included studies using the Newcastle–Ottawa scale for case-control studies are shown in Table 2. The risk of bias of each included study was low, receiving six to seven points. All studies clearly ascertained the LND status using surgical and pathology reports. The outcomes were generally reliably assessed using medical records with adequate follow-up duration. However, all studies had the potential for selection bias in assigning the case or control groups.

## 4. Effects of Interventions

### 4.1. CSS

Of the three studies that referred to CSS, there was no heterogeneity (*I*^2^ = 0%, Chi^2^ = 1.40, *p* = 0.50). As shown in Figure 2a, the pooled HR of these studies revealed that higher numbers of LNs removed during RNUx were associated with improved CSS in patients with UTUC regardless of pN status (HR = 0.95, 95% CI: 0.91–0.99; *p* = 0.07) (Figure 2a).

### 4.2. OS

There was no significant heterogeneity in the three studies that focused on OS (*I*^2^ = 26%, Chi^2^ = 2.70, *p* = 0.26). The data showed no significant difference in OS between the number of LNs removed (HR = 0.97, 95% CI: 0.94–1.01; *p* = 0.16) (Figure 2b).

### 4.3. PFS

Data for PFS were reported in only one study. The HR for RFS was 1.00 (95% CI: 0.48–2.07), indicating that the number of LNs removed during RNUx was not associated with recurrence rates in patients with UTUC (Figure 2c).

### 4.4. According to pN Stage

Three studies examined the association of the number of LNs removed with survival outcomes among patients with pN0 or pN+ stage disease. Higher numbers of LNs removed were associated with improved OS in pN0 patients (Figure 2d), and there was no difference in either CSS or RFS. In the pN+ patients, the number of LNs removed had no CSS, OS, or RFS benefits.

### 4.5. Publication Bias

Publication bias was judged from funnel plots. Visually symmetrical funnel plots suggested that the meta-analysis results had no obvious bias in any of the three categories (Figure 2).

## 5. Discussion

In this systematic review and meta-analysis incorporating six studies based on 5071 patients with UTUC, we defined the role of LND in UTUC patients. The meta-analysis revealed that higher numbers of LN removed during RNUx seemed to be associated with significantly improved oncologic outcomes. CSS was improved in the pooled results of the three studies, and OS was improved in the pooled results of the three studies involving specific pN0 patients. However, unfortunately, only a maximum of three studies were included in each survival outcomes. In addition, there were no randomized controlled trial (RCT) contained in this meta-analysis. This may suggest that a meta-analysis is likely premature at this point. While premature, this may be the best attempt at a meta-analysis on this subject with the available data at the moment. These results require confirmation in further high-quality trials with prospective and randomized setting.

Meticulous LND during RNUx may have a therapeutic role in the eradication of lymphatic micro-metastases and a staging role providing accurate staging and enabling postoperative risk stratification for patient counseling [8]. One retrospective study of 132 consecutive monocentric patients with muscle-invasive UTUC reported that the five year CSS was 67% versus 40% in patients undergoing LND or not, respectively [4]. In addition, LND emerged as an independent predictor of increased CSS in multivariate analysis when compared to no LND [4]. One interesting study demonstrated the effect of loco-regional oncologic control and reported that most regional LN recurrences occurred outside the dissected sites after LND for renal pelvis UTUC [27]. However, there is still a lack of high-level evidence for the survival value of LND for UTUC.

In this study, we confirmed that the number of LNs removed during LND was a major factor in the oncological outcomes for UTUC patients. Because the lymphatic drainage patterns of the renal pelvis and ureter vary considerably, anatomical templates of LND for UTUC are not standardized and are commonly left to the discretion of the treating surgeon [8]. Therefore, the number of LNs removed could be a surrogate to determine the adequate extent and the overall quality of the LND procedure [24]. International data from 13 institutions including 551 patients showed that 13 LNs needed to be removed and examined to achieve a 90% probability of detecting LN involvement, whereas the removal of at least eight LNs resulted in a 75% probability of detecting one or more positive LN [28]. In our meta-analysis, removing a higher number of LN improved the CCS by about 5%.

According to a study using the National Cancer Data Base in the United States, LND is only implemented in less than about 20% of the RNUx procedures [26]. Feared morbidity might be, at least in part, the reason most surgeons hesitate to perform LNDs [8]. However, perioperative mortality and morbidity due to LND are not serious, reported to be 0% and 14.2% (Clavien–Dindo classification ≥ grade 3 = 3.9%), respectively [3]. In another study, only chylous lymphatic leaks were directly attributable to the extended LND component of the procedure, occurring in only one of 19 patients [29]. In addition, removal of more LNs did not significantly prolong the operation time and was not associated with higher estimated blood loss or grade three or higher complications at 30 days [23]. Fortunately, as the use of robotic surgery is increasing, the frequency of LNDs for UTUC is also increasing [10]. We suggest that more extensive LND can be safely and efficiently performed in addition to the RNUx procedure.

Higher numbers of LN removed in N0 patients, but not in N+ patients, had a benefit for OS in our study. A similar result was found in a corollary multi-institutional consortium study of 552 patients, where higher nodal yield carried a protective effect in the pN0 cohort alone [28]. Preoperative assessment of lymph node status seems difficult, as clinicians do not have available imaging, cytology, or pathology techniques to assess lymph node invasion preoperatively in UTUC [30,31]. LND has long been the gold standard for LN staging, which is a key determinant of prognosis [26]. Among patients who are classified as pN0, higher numbers of LNs removed and examined decreased the potential for misclassification of those individuals with LN metastases as pN0 [26]. A more accurate diagnosis of the N stage could help to risk-stratify patients for better postoperative counseling and clinical decision-making [8]. In addition, more LNs removed may be truly therapeutic with the locoregional eradication of unidentified micro-foci of metastatic disease that could be missed on routine pathologic examination, ultimately resulting in improved oncologic outcomes [23,26].

## 6. Conclusions

Based on this updated systematic review and meta-analysis, higher numbers of LNs removed during RNUx seemed to be associated with improved survival outcomes in patients with UTUC. This study suggested that LND also may have oncological benefits in UTUC patients. Although still a controversial topic, meticulous LND should be considered, and efforts may have to be made to eradicate as many LNs as possible when administering RNUx for UTUC, especially in patients without clear evidence of LN metastasis. These results require confirmation in further high-quality trials.

## Figures and Tables

**Figure 1 jcm-09-01933-f001:**
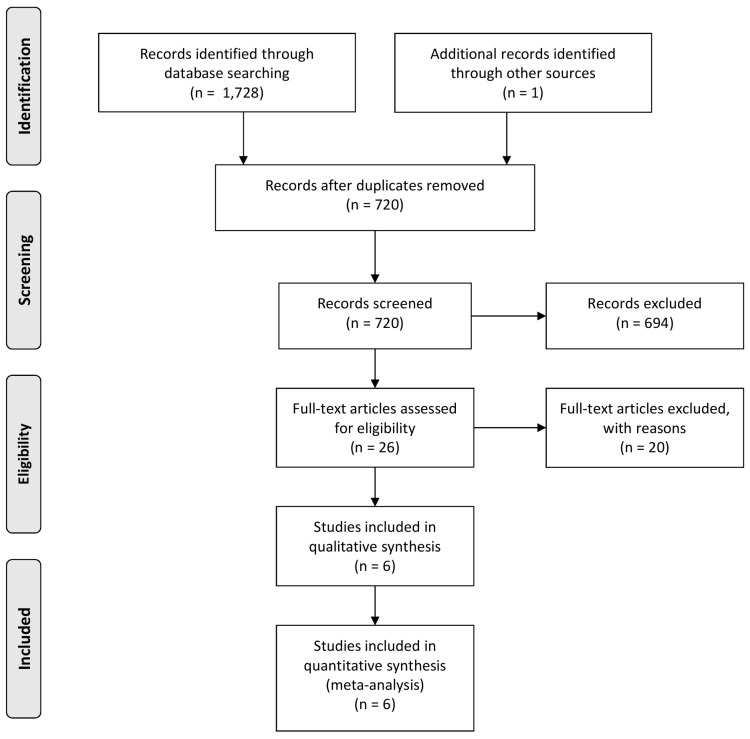
A flow chart showing literature search approach used in this meta-analysis.

**Figure 2 jcm-09-01933-f002:**
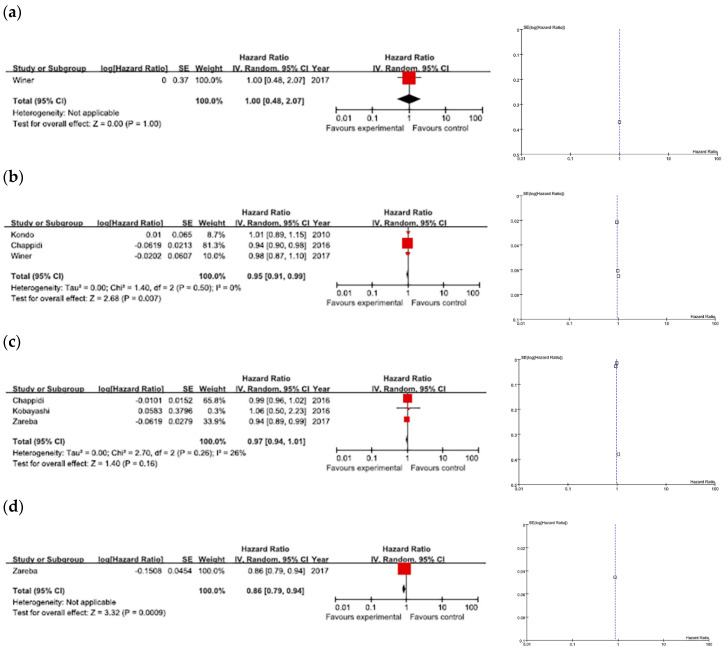
Forest plots of survival outcomes according to the number of lymph nodes removed, and funnel plots. (**a**) Cancer-specific survival, (**b**) overall survival, (**c**) progression-free survival, and (**d**) overall survival among N0 patients.

**Table 1 jcm-09-01933-t001:** Characteristics of the eligible studies.

Study	Years	Design	*N* of Patients	Stage	Median Follow-Up	Median Age	*N* Stage (% of N+)	Median *N* of LNs Removed (Mean)		Reports
				T1-2/T3-4			N0/N+	compLND	incompLND	
Kondo T et al. [24]	2010	Retro	119	68/51	60.5 (mean)	66.8 (mean)	98/21 (17.6)	9 (10.8)	4 (4.4)	CSS
Fajkovic H et al. [22]	2012	Multicenter, retro	817	-	33 (45.8, mean)	-	595/222 (27.2)	6	2	PFS, CSS
Kobayashi K et al. [25]	2015	Retro	54	10/45	33	69	14/40 (13.2)	8.5	5	OS
Chappidi MR et al. [10]	2016	Retro (SEER data)	721	309/410	28	-	466/255 (31.2)	5	1	CSS, OS
Zareba P et al. [26]	2017	Retro	2918	1433/1485	44	71	2151/767 (26.3)	Per 5 LNs	-	OS
Winer AG et al. [23]	2017	Retro	442	303/139	42	70	-	Per 5 LNs	-	PFS, CSS

*N* = number; LN = lymph node; compLND = complete lymphadenectomy; incompLND = incomplete lymphadenectomy; Retro = retrospective; CSS = cancer specific survival; PFS = progression free survival; OS = overall survival.

**Table 2 jcm-09-01933-t002:** Assessment of the quality of the non-radomized controlled trial (RCT) studies using the Newcastle–Ottawa scale for case-control studies.

	Selection				Comparability	Exposure			
Reference	CDA	RC	SC	DC	CCC	AE	SMA	NRR	Score
Kondo et al. [24]	☆			☆	☆☆	☆	☆		6
Fajkovic et al. [22]	☆	☆		☆	☆☆	☆	☆		7
Kobayashi et al. [25]	☆	☆		☆	☆☆	☆	☆		7
Chappidi et al. [10]	☆			☆	☆☆	☆	☆		6
Zareba et al. [26]	☆			☆	☆☆	☆	☆		6
Winer et al. [23]	☆			☆	☆☆	☆	☆		6

CDA: case definition adequate; RC: representativeness of the cases; SC: selection of controls; DC: definition of controls; CCC: comparability of cases and controls on the basis of the design or analysis; AE: ascertainment of exposure; SMA: same method of ascertainment for the cases and controls; NRR: non-response rate; A study can be awarded a maximum of one star (☆) for each numbered item within the selection and exposure categories. A maximum of two stars can be given for comparability.

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
