# Peer review of "Survival Benefits Based on the Number of Lymph Nodes Removed during Radical Nephroureterectomy for Upper Tract Urothelial Carcinoma: Systematic Review and Meta-Analysis"

_jcm, 2020, doi:10.3390/jcm9061933_

Round 1

Reviewer 1 Report

This is a really well written meta analysis of lymphadenectomy with UTUC. The title and abstract are very clear and appropriate. The methodology is entirely appropriate, and the whole paper is very clear and presented well.

Minor point page 4 line 148 ..."obviously different from wanted paper". This sentence needs correcting.

Author Response

reviewer 1.
This is a really well written meta analysis of lymphadenectomy with UTUC. The title and abstract are very clear and appropriate. The methodology is entirely appropriate, and the whole paper is very clear and presented well.

Minor point page 4 line 148 ..."obviously different from wanted paper". This sentence needs correcting.

reply>
Thank you for your sharp comment. I have corrected the manuscript of that part correctly. I apologize to you for my mistake.

Reviewer 2 Report

Concise yet very nice meta-analysis and written report with extensive and comprehensive search of the literature and a thorough statistical methodology (although I am not the expert in statistics of meta-analyses). I enjoyed reading it. 

Few comments:

-minor english mistakes in introduction in lines 46, 60-61, and in the sentence in lines 66-68 which disrupts the flow.

-in section "CSS" starting in line 175, please indicate whether higher number of LN removed improved CSS regardless of pN status.

-in section "According to pN stage", authors mention that in pN0 more LNs mean better OS and in pN+, number of LN did not correlate with any improved outcomes. Are the authors able to look at cN0 or cN+ as this will be the indicator for the surgeon on whether performing a LN dissection would be helpful.

-Discussion has a nice flow of ideas. In first paragraph, I would be careful with the wording "significantly improved oncologic outcomes" and would discuss the clinical relevance of HR of 0.97. Furthermore, distinguish the relevance in pN0 vs pN+, and maybe useful to add a reference on the correlation between cN and pN status in UTUC. 

-in line 40 of the discussion, authors say morbidity are not serious according to reference [3]. Please mention what those morbidities and their grades were.

-in conclusion, I would indicate that the benefit is minor (HR 0.97) and would change the choice of wording for "eliminate as many LN" and "administering RNUx"

Author Response

reviewer 2.
Concise yet very nice meta-analysis and written report with extensive and comprehensive search of the literature and a thorough statistical methodology (although I am not the expert in statistics of meta-analyses). I enjoyed reading it.

Few comments:

-minor english mistakes in introduction in lines 46, 60-61, and in the sentence in lines 66-68 which disrupts the flow.

reply>
Thank you for your sharp comment. It is all my mistakes. I have corrected those parts correctly in the manuscript. I apologize to you for my mistakes.

-in section "CSS" starting in line 175, please indicate whether higher number of LN removed improved CSS regardless of pN status.

reply>
Thank you for your valuable comments. I have attached a further explanation in the result section of the manuscript. It seems to be a more understandable manuscript because of your advice.

-in section "According to pN stage", authors mention that in pN0 more LNs mean better OS and in pN+, number of LN did not correlate with any improved outcomes. Are the authors able to look at cN0 or cN+ as this will be the indicator for the surgeon on whether performing a LN dissection would be helpful.

reply>
Thank you for your valuable comments. I totally agree with your opinion. An accurate clinical N stage will be of great help in determining whether LND is applied and the extent of LND during surgery. However, there are still few methods to accurately assess the n stage preoperatively, and the gold standard is still considered a surgical LND. I have added content to this in the discussion.

-Discussion has a nice flow of ideas. In first paragraph, I would be careful with the wording "significantly improved oncologic outcomes" and would discuss the clinical relevance of HR of 0.97. Furthermore, distinguish the relevance in pN0 vs pN+, and maybe useful to add a reference on the correlation between cN and pN status in UTUC.

reply>
Thank you for your sharp comment. I totally agree with your opinion.
Readers may be misunderstood in situations where the survival gain after LND is not very large. I revised the content as your points.

-in line 40 of the discussion, authors say morbidity are not serious according to reference [3]. Please mention what those morbidities and their grades were.
reply>
Thank you so much for your valuable comment. As you pointed out, displaying the correct grade or ratio will be a more accurate expression. I added a grade of complication according to Clavien-Dindo classification in the manuscript.

-in conclusion, I would indicate that the benefit is minor (HR 0.97) and would change the choice of wording for "eliminate as many LN" and "administering RNUx"
reply>
Thank you for your sharp comment. I totally agree with your opinion, again.
I have revised the content so that the readers do not misunderstand them. Thank you again for your sincere advice.

Reviewer 3 Report

The authors present a well-conducted systematic review and meta-analysis of the survival effect of lymphadenectomy during radical nephroureterectomy. As the authors acknowledge, very few studies give complete data on the lymph node yield, lymph node positivity, and survival after nephroureterectomy.

Unfortunately, only a maximum of 3 studies are included in each Forest plot of survival outcomes. Only 1 study is included for the Forest plot of cancer-specific survival and overall survival. This suggests that a meta-analysis is likely premature at this point. While premature, this may be the best attempt at a meta-analysis on this subject with the available data.

Author Response

reviewer 3.
The authors present a well-conducted systematic review and meta-analysis of the survival effect of lymphadenectomy during radical nephroureterectomy. As the authors acknowledge, very few studies give complete data on the lymph node yield, lymph node positivity, and survival after nephroureterectomy.

Unfortunately, only a maximum of 3 studies is included in each Forest plot of survival outcomes. Only 1 study is included for the Forest plot of cancer-specific survival and overall survival. This suggests that a meta-analysis is likely premature at this point. While premature, this may be the best attempt at a meta-analysis on this subject with the available data.
reply>
Thank you for your sharp comment. I totally agree with your opinion.
Readers may be misunderstood in situations where the survival gain after LND is not very large. I revised the content as your points.

Reviewer 4 Report

Dear authors,

Congratulations for writing this interesting meta-analysis.

Care should be taken when interpreting the results since only retrospective studies were included and the mean number of removed lymph nodes is rather low. Therefore, it should be recommended that prospective randomized studies should be performed.

Author Response

reviewer 4.
Dear authors,

Congratulations for writing this interesting meta-analysis.

Care should be taken when interpreting the results since only retrospective studies were included and the mean number of removed lymph nodes is rather low. Therefore, it should be recommended that prospective randomized studies should be performed.
reply>
Thank you for your sharp comment. I totally agree with your opinion.
Readers may be misunderstood in situations where the survival gain after LND is not very large. I revised the content as your points.